# Synergism of a Novel 1,2,4-oxadiazole-containing Derivative with Oxacillin against Methicillin-Resistant *Staphylococcus aureus*

**DOI:** 10.3390/antibiotics10101258

**Published:** 2021-10-16

**Authors:** Elisabetta Buommino, Simona De Marino, Martina Sciarretta, Marialuisa Piccolo, Carmen Festa, Maria Valeria D’Auria

**Affiliations:** Department of Pharmacy, University of Naples Federico II, Via Montesano 49, 80131 Naples, Italy; elisabetta.buommino@unina.it (E.B.); simona.demarino@unina.it (S.D.M.); martina.sciarretta@unina.it (M.S.); marialuisa.piccolo@unina.it (M.P.); madauria@unina.it (M.V.D.)

**Keywords:** 1,2,4-oxadiazole, *Staphylococcus aureus*, MRSA, antimicrobial activity, synergistic interaction, structure–activity relationship, drug discovery

## Abstract

*Staphylococcus**aureus* is an important opportunistic pathogen that causes many infections in humans and animals. The inappropriate use of antibiotics has favored the diffusion of methicillin-resistant *S. aureus (*MRSA), nullifying the efforts undertaken in the discovery of antimicrobial agents. Oxadiazole heterocycles represent privileged scaffolds for the development of new drugs because of their unique bioisosteric properties, easy synthesis, and therapeutic potential. A vast number of oxadiazole-containing derivatives have been discovered as potent antibacterial agents against multidrug-resistant MRSA strains. Here, we investigate the ability of a new library of oxadiazoles to contrast the growth of Gram-positive and Gram-negative strains. The strongest antimicrobial activity was obtained with compounds **3** (4 µM) and **12** (2 µM). Compound **12**, selected for further evaluation, was found to be noncytotoxic on the HaCaT cell line up to 25 µM, bactericidal, and was able to improve the activity of oxacillin against the MRSA. The highest synergistic interaction was obtained with the combination values of 0.78 μM for compound **12,** and 0.06 μg/mL for oxacillin. The FIC index value of 0.396 confirms the synergistic effect of compound **12** and oxacillin. MRSA treatment with compound **12** reduced the expression of genes included in the *mec* operon. In conclusion, **12** inhibited the growth of the MRSA and restored the activity of oxacillin, thus resulting in a promising compound in the treatment of MRSA infection.

## 1. Introduction

Antibiotics are a class of drugs traditionally used as the first strategy (weapon) against serious infections caused by bacteria. Among these, *Staphylococcus aureus* is an important opportunistic pathogen that causes many infections in humans and animals, such as skin and lung infections, endocarditis, toxic shock syndrome, and sepsis [1]. It is a member of the human microbiota (about 30% of the human population is colonized with *S. aureus*) with a pathogenic potential derived from the ability to express a wide variety of virulence factors used to overcome the host defense mechanism [2]. Antibiotics have generally shown efficacy in the treatment of bacterial infections, but their inappropriate and indiscriminate use has favored the diffusion of resistant bacteria. This is the case with β-lactams and the selection of methicillin-resistant *S. aureus* (MRSA), which has nullified the efforts undertaken in the discovery of antimicrobial agents [3]. *S. aureus* can also form biofilm on medical devices, and this represents a further complication in the management of infections because of the difficulties of antibiotics to penetrate the biofilm layer [4]. For all these reasons, *S. aureus* is considered one of the major causes of hospital-acquired infections worldwide [5].

The discovery of new targets or compounds that are potentially able to control diseases by resistant bacteria represents an urgent goal [6].

The 1,2,4-oxadiazole heterocyclic ring represents an important scaffold in organic and medicinal chemistry, with an elevated versatility that affords molecular diversity.

A vast number of 1,2,4-oxadiazole-bearing molecules have been found to possess diverse biological activities, such as anticancer, anti-inflammatory, anticonvulsant, antiviral, antibacterial, and anti-Alzheimer activities, as well as specific inhibitory properties against enzymes [7,8,9,10].

In 2014, using in silico screening, 1,2,4-oxadiazole-derived compounds were discovered by Chang and coworkers as a new class of non-β-lactam antibiotics [11].

On the basis of the previously disclosed lead compound **1**, our study aims to further expand the structural diversity around the 1,2,4-oxadiazole-selected scaffold in order to identify novel and promising antibacterial chemical entities.

New 1,2,4-oxadiazoles (**3**–**17**) were designed by fixing a common 4-indole ring at C-5, and by varying the aromatic substituent in position 3. These compounds were synthesized based on amidoxime and carboxylic acid heterocyclization and were tested for their antimicrobial activity against *S. aureus* 29213 and 43300 (MRSA), *P. aeruginosa,* and *K. pneumoniae*.

## 2. Results

### 2.1. Compounds and Chemistry

We report herein the synthesis and SAR results, against *S. aureus,* of a library of fifteen synthetic oxadiazoles (**3**–**17**), featuring a common 5-(4′ indolyl) substituent, and different aromatic substitutions at C-3 (Figure 1).

Starting from the lead structure **1** (Figure 2) discovered by Chang and coworkers [11], and featuring four interconnected aromatic rings named **A**-**D**, the same authors report an extensive structure–activity relationship that discloses compounds bearing a 5-indole ring A, for example, compound **2**, (Figure 2), as the best candidates, in terms of activity, against MSRA strains, low toxicity, oral bioavailability, and in vivo activity in murine models of infection [12,13,14].

Very recently, in the frame of our interest toward the development of 1,2,4-oxadiazole-based active molecules [15,16], we designed and synthesized a small library of oxadiazole-containing compounds endowed with a polypharmacological profile against the enzymes of the eicosanoid biosynthesis [17].

Since all of the compounds within the library were filtered through the detailed in silico assessment of their drug-like properties, we envisaged a repositioning of the molecules that were lacking in anti-inflammatory activity.

Compound **3**, synthetized in our previous study [17], features some analogies with compounds **1** and **2**, namely, the concatenation of four aromatic rings and a regioisomeric 4-indole group at the C-5 position of the 1,2,4-oxadiazole ring (Figure 2). On the basis of these observations, we analyzed the antimicrobial activity of compound **3** on two strains of *S. aureus* (ATCC 29213 and MRSA ATCC 43300). As reported in Table 1, it showed strong activity, with an MIC value of 4 µM, on both *S. aureus* strains.

Thus, starting from this encouraging finding, we further expanded the chemodiversity around the 1,2,4-oxadiazole ring by fixing the 4-indolyl system as ring A, and by varying the structures of rings C and D. In particular, in some compounds, we removed ring D and added different substituents on ring C (see compounds **5**–**9**), while in others, we replaced the diphenyl ether portion (rings C and D) with different benzyl, biphenyl, thienyl, or benzyloxyphenyl rings substituted at different positions (see compounds **3**,**4** and **10**–**17**) (Figure 1). A key attractive feature of this class of compounds is its easy synthetic preparation, outlined in Figure 1. In particular, indole-4-carboxylic acid was reacted, using HBTU as a coupling agent, with amidoximes **2a**-**o**, in turn, prepared by the reaction of corresponding nitriles **1a**-**o** with hydroxylamine hydrochloride to give compounds **3**–**17**. While most of these nitriles are commercially available, nitriles **1n** and **1o** had to be synthesized following Williamson reactions between 4-cyanophenol and methyl 4-(bromomethyl)benzoate, or methyl 3-(bromomethyl)benzoate, respectively (procedures reported in Materials and Methods).

The structure of the synthetized compounds was confirmed by analysis of the NMR data (^1^H and^13^C NMR spectra) and ESI-MS (see Materials and Methods).

### 2.2. Antimicrobial Activity

The new synthetized compounds (**4**–**17**) were tested for their antimicrobial activity against *S. aureus* ATCC 29213 and *S. aureus* ATCC 43300, and two Gram-negative strains, *P. aeruginosa* ATCC 27853 and *K. pneumoniae* ATCC 13883. The results are reported in Table 1. Compounds **6**, and **11**–**14** showed the strongest antimicrobial activity against the two Staphylococcal strains, reporting MIC values of 6.25 µM. However, by further lowering the concentration, a more accurate estimate of the MIC value was obtained for compound **12** (2 µM). Compound **7** resulted active at 100 µM, while compounds **4**, **8**, **10**, **16** and **17** were not active. None of the compounds were active against *P. aeruginosa* ATCC 27853 and *K. pneumoniae* (data not shown).

The minimum bactericidal concentration (MBC) resulted in 2 μM for compound **12,** and 4 μM for compound **3**. To confirm the bacteriostatic, or bactericidal activity, of **12,** we performed the time kill assay on *S. aureus* ATCC 29213 and *S. aureus* ATCC 43300 at the MIC value (Figure 3) [18]. Compound **12** caused an over 3log_10_ -fold reduction in the bacterial count after 3 h, compared to initial inoculum, and completely abrogated the cell growth 24 h after.

### 2.3. Synergistic Study

The synergism between **12** and oxacillin against MRSA was determined using the checkerboard technique. The highest synergistic interaction was obtained with the combination values of 0.78 μM for compound **12,** and 0.06 μg/mL for oxacillin. The FIC index value of 0.396 confirmed the synergistic effect of compound **12** and oxacillin.

### 2.4. Molecular Analysis

Here we investigated the effect of the compound **12** treatment on *mecA*, *mec1*, *mecR1* gene expression modulation [19]. The treatment with the subinhibitory concentrations of **12** (1 µM and 0.39 μM), for 30 min, induced a reduced expression of all the *mec* operon genes, whereas oxacillin strongly induced them both at subinhibitory (0.03 µg/mL) and MIC values (10 µg/mL) (Figure 4). On the contrary, the synergic interaction between **12** and oxacillin, used at subinhibitory concentrations, counteracted the effect of oxacillin, restoring the levels of expression of all genes (*mecA*, *mec1*, and *mecR1*) to values comparable to **12**-treated cells.

### 2.5. Cytotoxicity Studies

The cytotoxic properties of **12** were investigated using HaCaT keratinocytes as human model in vitro. The results show that compound **12** does not interfere with cell viability at lower concentrations (≤25 µM) after 48 h of incubation, while the cytotoxic activity of **12** on the HaCaT cell line at higher concentrations (≥35) was observed. The concentration–effect curve (Figure 5a)—here reported in terms of a “cell survival index” that combines the measurements of cell number and viability—shows a typical concentration-dependent sigmoid trend yielding IC_50_ values in the low micromolar range (Figure 5b). In order to exclude any effect given by the vehicle, DMSO was used as negative control. As reported in Figure 5a, there is no interfering with cell viability following incubation with DMSO, even at higher concentrations.

## 3. Discussion

The oxadiazoles are a class of five-membered heterocycles that are of considerable interest in organic and medicinal chemistry [20]. In some bioactive compounds, they act as bioisosteres of ester and amide, with improved pharmacokinetic and pharmacodynamic profiles and enhanced selectivity; in other cases, the rigid heterocyclic ring acts as a flat aromatic spacer, assuring the appropriate substituent geometry. A vast number of oxadiazole-containing scaffolds are found to possess diverse biological activities, such as anticancer, anti-inflammatory, antiallergic, anticonvulsant, antiviral, antibacterial, anti-Alzheimer activities, as well as specific inhibitor properties against enzymes [21,22].

This study aims to further explore the chemical space around the oxadiazole ring. In particular, we have focused our attention on 1,2,4-oxadiazole-containing derivatives in order to find easily accessible chemical entities active against clinically relevant bacterial strains.

Starting from the lead compound **1**, obtained by Chang and coworkers after an extensive SAR investigation of oxadiazole derivatives [11], and taking into account the promising MIC of the related structure **3**, in the present study, we fixed the 4-indolyl system as ring A, and we varied the size and the functionalization of the aromatic right portion at the C-3 position of the oxadiazole ring. The SAR was established by an initial screening of the synthetic derivatives against two MSRA *S. aureus* strains. Among the fifteen synthesized compounds, two (**3** and **12**) displayed the best activity, and **12** was selected as the most promising for its ability to contrast the growth of *S. aureus* and the MRSA strain at an MIC value of 2 µM.

The MIC values reported were indicative of the marked influence of the substituents at C-3 of the oxadiazole ring on the antimicrobial activity. In particular, the most active compounds, **3** and **12**, featured, respectively, 4-(3-thienyl)phenyl and 1,1′-biphenyl-4-yl substituents, namely, a *para*-biaryl unsubstituted system. No activity was observed for compound **4,** with ring C *meta*-substituted with 3-thienyl ring. Moreover, the weaker activity of compound **3** (MIC 4 µM), as compared to compound **12** (MIC 2 µM), could be ascribed to the lesser degree of aromaticity of the heterocyclic thiophene in compound **3** vs. the phenyl moiety in compound **12**.

The introduction of hydroxy groups on ring C, and the simultaneous absence of ring D in compounds **7** and **8,** or the introduction of a benzyl group (**10**), dramatically decreased the antibacterial activity, resulting in MIC values ≥ 100 µM. The presence of the OH group at the *orto* position of ring C (compound **9**) enhanced activity with an MIC of 25 µM. On the other hand, the strong electron-withdrawing group, CF_3_, enhanced the activity (**6** vs. **5** and **11** vs. **10**). The introduction of an oxymethylene ether spacer between rings C and D (compounds **16** and **17**) caused the loss of antibacterial activity. Compounds **13** and **14,** characterized by the introduction of hydroxy groups on the biphenyl ring, showed moderate activity, with an MIC of 6.25 µM, while compound **15** showed weaker activity (MIC 12.5 µM).

Of interest, previous studies have reported that this class of compounds shows antimicrobial activity against *S. aureus* at very low concentrations (values ranging from 1 to 2 µg/mL) [13,23]. However, compound **12** resulted in activity at a lower concentration (2 µM corresponds to 0.674 µg/mL), representing the most active compound so far developed within the oxadiazole class. Therefore, the conjugated *para*-biphenyl system in **12** appears more effective than the *para*-diphenyl ether moiety in compounds **1** and **2**. Moreover, compound **12** was not toxic at concentrations less than 25 µM, and strongly contrasted the effect of oxacillin when used in combination. The FIC index demonstrated the synergic activity of **12** with oxacillin. The latter result is of interest since antibiotic research worldwide is focused on the discovery of new compounds that act synergically with the old class of unutilized antibiotic, because of the lack of activity against multidrug-resistant bacteria, but which are well-known for their pharmacokinetic and safety parameters [24]. Repurposing existing antibiotics would immediately respond to clinical requests, as these drugs have long been administered and are already approved for human use. In this context, programs and interventions aiming at optimizing the use of antimicrobial drugs, termed “antimicrobial stewardship”, have been welcome [25]. The clinical resistance to β-lactam antibiotics by MRSA occurs because of the acquisition of the *mecA* gene, contained in the staphylococcal cassette chromosome mec (SCCmec), which encodes a cell-wall DD-transpeptidase that is not inhibited by the latter antibiotics [26,27]. Bacteria are, thus, not deprived of the final stages of cell-wall biosynthesis, and they can grow and replicate undisturbed.

The regulation of *mecA* expression is controlled by its own regulators, *mecR1* and *mecI*, also carried on the SCCmec element. *MecA* transcription is so repressed by *mec1*, and only the presence of β-lactam antibiotics, detected by the sensory domains in *MecR1*, removes the repression of *mecA* transcription by MecI, leading to *mecA* transcription, PBP2a production, and the expression of methicillin resistance [28]. The results here demonstrate that the treatment of MRSA with compound **12** induces a decreased expression of the genes included in the *mec* operon. More interestingly, **12** contrasts the protective response of bacteria to oxacillin treatment, namely, the strong induction of the *mecA* gene and the related genes of the operon, resensitizing MRSA to beta-lactam antibiotics. The ability of oxadiazole to inhibit PBP2a is known [10]. We can speculate that compound **12** displays the same effect, inhibiting the activity of the enzyme. It can, perhaps, bind to the *MecR1* domain competing with oxacillin binding, thus inhibiting the transcription of the *mec* operon genes. Further studies are necessary in order to ascertain the hypothesis here formulated.

Furthermore, cytotoxicity studies performed by bioscreens in vitro revealed an activity of **12** in inhibiting cell growth and proliferation only at higher concentrations.

## 4. Materials and Methods

### 4.1. Materials

All chemical reagents are commercially available from Sigma Aldrich^®^ (Milan, Italy) and TCI (Tokyo, Japan). Solvents and reagents were used as supplied from commercial sources, with the exception of methanol, which was anhydrified from magnesium methoxide as follows: Magnesium turnings (5 g) and iodine (0.5 g) were refluxed in a small amount of methanol (50–100 mL), until all the magnesium was reacted. The mixture was diluted (up to 1 L) with methanol, refluxed for 2–3 h, and then distilled under argon. All reactions were carried out under argon atmosphere using flame-dried glassware.

The reaction progress was monitored via thin-layer chromatography (TLC) on Alugram silica gel G/UV254 plates. The C-18 Phenomenex Luna column was used as a chromatography column. The silica gel, MN Kiesel gel 60 (70–230 mesh) of Macherey-Nagel, was used for flash chromatography. HPLC was performed using a Waters Model 510 pump equipped with a Waters Rheodyne injector and a differential refractometer, model 401.The purities of the compounds were estimated to be greater than 95% by HPLC.

NMR spectra were recorded on Bruker Avance NEO 400 and 700 spectrometers equipped with an RT-DR-BF/1H-5 mm-OZ SmartProbe (^1^H at 400 MHz and ^13^C at 100 MHz; ^1^H at 700 MHz and ^13^C at 175 MHz).

Coupling constants (*J*) are given in Hertz (Hz), chemical shifts were reported in δ (ppm), and referred to the residual CH_3_OD and CHCl_3_ as internal standards (δ_H_ = 3.31 and δ_C_ = 49.0 ppm; δ_H_ = 7.26 and δ_C_ = 77.0 ppm). All the recorded signals were in accordance with the proposed structures. Spin multiplicities are given as s (singlet), br s (broad singlet), d (doublet), or m (multiplet). ESI-MS analysis was carried out on a mass spectrometer, LTQ-XL.

### 4.2. Synthesis

#### 4.2.1. Synthetic Procedures for Nitriles **1n** and **1o**

Methyl 4-(bromomethyl)benzoate and methyl 3-(bromomethyl)benzoate (1.2 mol eq.) were added to a solution of 4-cyanophenol (1 mol eq.) and K_2_CO_3_ (2 mol eq.), respectively, in DMF dried at 100 °C. The reaction was stirred at 25 °C overnight, then fractionated in water and ethyl acetate three times. The organic layers were dried over anhydrous Na_2_SO_4_, and filtrated and concentrated under vacuum to obtain the correspondent nitriles, methyl 4-[(4-cyanobenzyl)oxy]benzoate (**1n**) and methyl 3-[(4-cyanobenzyl)oxy]benzoate (**1o**).

#### 4.2.2. General Synthetic Procedures for Amidoximes **2a**-**o**

Each nitrile (1 mol eq.) was dissolved in dry methanol and potassium carbonate (1.5 mol eq.), and hydroxylamine hydrochloride (2.5 mol eq.) was added to the solution. The mixture was refluxed for 8 h in inert atmosphere. The reaction was concentrated under vacuum to remove MeOH, diluted with water, and extracted three times with CH_2_Cl_2_. The organic phases were dried over anhydrous Na_2_SO_4_, filtered, and concentrated under vacuum to obtain the correspondent amidoximes (yield 70–90%). The product was subjected to the next steps without any purification.

#### 4.2.3. Synthetic procedures for compounds **3**–**17**

Indole-4-carboxylic acid (1.2 mol eq.) was dissolved in dry DMF and DIPEA (1.8 mol eq.), and the coupling reagent, HBTU (1.5 mol eq.), was added to the solution. Amidoximes **2a**-**o** (1 mol eq.) were added ten minutes later in the corresponding reactions. The mixture was stirred at 140° C for 12 h, then fractionated in water and ethyl acetate three times. The organic layers were cooled and washed three times with a saturated solution of LiBr, and then with a saturated solution of NaHCO_3_ and distilled water. The organic layer was dried over anhydrous Na_2_SO_4_, filtered, and concentrated under reduced pressure obtaining the oxadiazoles **3**–**17**. The crude reactions were purified as follows:

**5-(*1H*-indol-4-yl)-3-(4-(3-thienyl)phenyl)-1,2,4-oxadiazole (3)****.** NMR data as previously reported [17].

**5-(*1H*-indol-4-yl)-3-(3-(3-thienyl)phenyl)-1,2,4-oxadiazole (4).** An analytic sample of crude reaction was purified by HPLC using a Luna Column C-18 (10 µm, 250 mm × 10 mm), with MeOH/H_2_O (80:20) as eluent (flow rate 3.00 mL/min), affording compound **4** (t_R_ = 9.0 min). ^1^H NMR (700 MHz, CD_3_OD): δ_H_ 8.20 (1H, s), 7.99 (1H, d, *J* = 7.4 Hz), 7.83 (1H, d, *J* = 7.8 Hz), 7.77 (1H, m), 7.75 (1H, d, *J* = 7.8 Hz), 7.69 (1H, d, *J* = 7.8 Hz), 7.58 (1H, dd, *J* = 5.0, 1.2 Hz), 7.51 (1H, d, *J* = 3.1 Hz), 7.45 (1H, d, *J* = 3.1 Hz), 7.25 (1H, t, *J* = 7.8 Hz) 7.13 (1H, d, *J* = 3.0 Hz); ^13^C NMR (175 MHz, CD_3_OD): δ_C_ 167.8, 160.0, 142.8, 138.7, 137.7, 133.6, 130.3, 129.8, 129.1, 128.4, 127.6, 127.3, 126.8, 126.4, 123.8, 122.2, 121.6, 121.1, 117.8, 103.5. ESI-MS *m/z* 344.2 [M + H]^+^.

**5-(*1H*-indol-4-yl)-3-phenyl-1,2,4-oxadiazole (5)**. An analytic sample of crude reaction was purified by HPLC using a Luna Column C-18 (10 µm, 250 mm × 10 mm), with MeOH/H_2_O (80:20) as eluent (flow rate 3.00 mL/min), affording compound **5** (t_R_ = 21.0 min). ^1^H NMR (700 MHz, CD_3_OD): δ_H_ 8.20 (2H, d, *J* = 7.7 Hz), 8.01 (1H, d, *J* = 7.4 Hz), 7.71 (1H, d, *J* = 7.8 Hz), 7.56 (3H, ovl), 7.51 (1H, d, *J* = 3.1 Hz), 7.31 (1H, t, *J* = 7.8 Hz), 7.28 (1H, d, *J* = 3.1 Hz); ^13^C NMR (175 MHz, CD_3_OD): δ_C_ 178.7, 170.1, 138.7, 132.5, 130.4 (3C), 128.9, 128.6 (2C), 127.6, 122.4, 122.2, 117.4, 116.1, 103.6. ESI-MS *m/z* 262.1 [M + H]^+^.

**5-(*1H*-indol-4-yl)-3-(4-(trifluoromethyl)phenyl)-1,2,4-oxadiazole (6).** An analytic sample of crude reaction was purified by HPLC using a Luna Column C-18 (10 µm, 250 mm × 10 mm), with MeOH/H_2_O (90:10) as eluent (flow rate 3.00 mL/min), affording compound **6** (t_R_ = 17.5 min). ^1^H NMR (700 MHz, CD_3_OD): δ_H_ 8.29 (2H, d, *J* = 8.0 Hz), 7.95 (1H, d, *J* = 7.7 Hz), 7.79 (2H, d, *J* = 8.0 Hz), 7.67 (1H, d, *J* = 7.7 Hz), 7.48 (1H, d, *J* = 3.0 Hz), 7.26 (1H, t, *J* = 7.8 Hz) 7.24 (1H, d, *J* = 3.0 Hz); ^13^C NMR (175 MHz, CD_3_OD): δ_C_ 178.5, 165.7, 138.5, 133.5 (q, *J* = 32.7 Hz), 132.2, 128.9 (2C), 128.7, 127.4, 126.8 (2C, q, *J*= 3.5 Hz), 125.4 (q, *J* = 271.2 Hz), 122.2, 121.9, 117.6, 115.3, 103.3. ESI-MS *m/z* 330.1 [M + H]^+^.

**5-(*1H*-indol-4-yl)-3-(4-(hydroxy)phenyl)-1,2,4-oxadiazole (7).** An analytic sample of crude reaction was purified by HPLC using a Luna Column C-18 (10 µm, 250 mm × 10 mm), with MeOH/H_2_O (80:20) as eluent (flow rate 3.00 mL/min), affording compound **7** (t_R_ = 10.0 min). ^1^H NMR (400 MHz, CD_3_OD): δ_H_ 8.03 (2H, d, *J* = 8.7 Hz), 7.99 (1H, d, *J* = 7.8 Hz), 7.70 (1H, d, *J* = 7.8 Hz), 7.51 (1H, d, *J* = 3.1 Hz), 7.30 (1H, t, *J* = 7.8 Hz) 7.26 (1H, d, *J* = 3.1 Hz), 6.94 (2H, d, *J* = 8.7 Hz); ^13^C NMR (100 MHz, CD_3_OD): δ_C_ 178.0, 169.8, 161.7, 138.5, 130.2 (2C), 130.1, 128.5, 127.4, 122.0 (2C), 119.5, 117.3, 116.7, 116.0, 103.35. ESI-MS *m/z* 278.1 [M + H]^+^.

**5-(*1H*-indol-4-yl)-3-(3-(hydroxy)phenyl)-1,2,4-oxadiazole (8).** An analytic sample of crude reaction was purified by HPLC using a Luna Column C-18 (10 µm, 250 mm × 10 mm), with MeOH/H_2_O (80:20) as eluent (flow rate 3.00 mL/min), affording compound **8** (t_R_ = 11.5 min). ^1^H NMR (700 MHz, CD_3_OD): δ_H_ 8.00 (1H, d, *J* = 7.6 Hz), 7.71 (1H, d, *J* = 8.0 Hz), 7.67 (1H, d, *J* = 7.6 Hz), 7.64 (1H, d, *J* = 1.6 Hz), 7.51 (1H, d, *J* = 3.0 Hz), 7.37 (1H, t, *J* = 8.0 Hz), 7.31 (1H, t, *J* = 7.6 Hz), 7.28 (1H, d, *J* = 3.0 Hz), 6.97 (1H, dd, *J* = 8.0, 1.6 Hz); ^13^C NMR (175 MHz, CD_3_OD): δ_C_ 178.3, 169.9, 159.2, 138.5, 131.1, 129.7, 128.6, 127.4, 122.1, 122.0, 119.6, 119.3, 117.4, 115.9, 115.1, 103.3. ESI-MS *m/z* 278.1 [M + H]^+^.

**5-(*1H*-indol-4-yl)-3-(2-(hydroxy)phenyl)-1,2,4-oxadiazole (9).** An analytic sample of crude reaction was purified by HPLC using a Luna Column C-18 (10 µm, 250 mm × 10 mm), with MeOH/H_2_O (80:20) as eluent (flow rate 3.00 mL/min), affording compound **9** (t_R_ = 19.0 min). ^1^H NMR (400 MHz, CD_3_OD): δ_H_ 8.14 (1H, d, *J* = 7.7 Hz), 8.05 (1H, d, *J* = 7.4 Hz), 7.76 (1H, d, *J* = 8.0 Hz), 7.56 (1H, d, *J* = 3.0 Hz), 7.45 (1H, t, *J* = 8.0 Hz), 7.35 (1H, t, *J* = 7.7 Hz), 7.22 (1H, d, *J* = 3.0 Hz), 7.07 (1H, d, *J* = 8.0 Hz) 7.05 (1H, d, *J* = 7.7 Hz); ^13^C NMR (100 MHz, CD_3_OD): δ_C_ 177.1, 168.6, 158.4, 138.6, 134.1, 129.6, 129.0, 127.3, 122.6, 122.0, 120.9, 118.4, 118.0, 115.1, 112.6, 103.0. ESI-MS *m/z* 278.1 [M + H]^+^.

**3-benzyl-5-(*1H*-indol-4-yl)-1,2,4-oxadiazole (10).** An analytic sample of crude reaction was purified by HPLC using a Luna Column C-18 (10 µm, 250 mm × 10 mm), with MeOH/H_2_O (80:20) as eluent (flow rate 3.00 mL/min), affording compound **10** (t_R_ = 13.5 min). ^1^H NMR (400 MHz, CD_3_OD): δ_H_ 7.92 (1H, dd, *J* = 7.5, 0.8 Hz), 7.70 (1H, d, *J* = 8.0 Hz), 7.49 (1H, d, *J* = 3.2 Hz), 7.43 (2H, d, *J* = 7.5 Hz), 7.36 (2H, t, *J* = 7.5 Hz), 7.33 (2H, ovl), 7.15 (1H, dd, *J* = 3.2, 0.8 Hz); ^13^C NMR (175 MHz, CD_3_OD): δ_C_ 178.5, 171.3, 138.6, 137.5, 130.2 (2C), 128.9 (2C), 128.7, 128.1, 127.4, 122.2, 122.0, 117.5, 115.9, 103.2, 33.0. ESI-MS *m/z* 276.1 [M + H]^+^.

**5-(*1H*-indol-4-yl)-3-(4-(trifluoromethyl)benzyl)-1,2,4-oxadiazole (11).** An analytic sample of crude reaction was purified by HPLC using a Luna Column C-18 (10 µm, 250 mm × 10 mm), with MeOH/H_2_O (80:20) as eluent (flow rate 3.00 mL/min), affording compound **11** (t_R_ = 18.0 min). ^1^H NMR (700 MHz, CD_3_OD): δ_H_ 7.90 (1H, d, *J* = 7.5 Hz), 7.68 (2H, d, *J* = 8.0 Hz), 7.65 (2H, d, *J* = 8.0 Hz), 7.61 (2H, d, *J* = 8.0 Hz), 7.47 (1H, d, *J* = 3.1 Hz), 7.26 (1H, t, *J* = 7.8 Hz) 7.13 (1H, d, *J* = 3.1 Hz), 4.28 (2H, s); ^13^C NMR (175 MHz, CD_3_OD): δ_C_ 178.6, 170.6, 142.0, 138.5, 130.8 (2C), 130.3 (q, *J* = 32.2 Hz), 128.7, 127.3, 127.0 (2C, q, *J* = 3.7 Hz), 126.1 (q, *J* = 271 Hz), 122.1, 121.9, 117.5, 115.7, 103.1, 32.9. ESI-MS *m/z* 344.1 [M + H]^+^.

**3-([1,1′-biphenyl]-4-yl)-5-(*1H*-indol-4-yl)-1,2,4-oxadiazole (12).** An analytic sample of crude reaction was purified by HPLC using a Luna Column C-18 (10 µm, 250 mm × 10 mm), with MeOH/H_2_O (80:20) as eluent (flow rate 3.00 mL/min), affording compound **12** (t_R_ = 10.0 min). ^1^H NMR (700 MHz, CD_3_OD): δ_H_ 8.27 (2H, d, *J* = 8.3 Hz), 8.03 (1H, d, *J* = 7.4 Hz), 7.82 (2H, d, *J* = 8.3 Hz), 7.71 (3H, ovl), 7.53 (1H, d, *J* = 3.0 Hz), 7.48 (2H, t, *J* = 7.4 Hz), 7.39 (1H, t, *J* = 7.4 Hz), 7.32 (1H, d, *J* = 7.4 Hz), 7.30 (1H, d, *J* = 3.1 Hz); ^13^C NMR (175 MHz, CD_3_OD): δ_C_ 178.4, 169.7, 145.3, 141.3, 138.4, 130.0 (2C), 129.0, 128.9 (2C), 128.5, 128.4 (2C), 128.0 (2C), 127.3 (2C), 122.1, 121.9,117.4, 115.8, 103.3. ESI-MS *m/z* 338.2 [M + H]^+^.

**3-(4′-hydroxy-[1,1′-biphenyl]-4-yl)-5-(*1H*-indol-4-yl)-1,2,4-oxadiazole (13).** An analytic sample of crude reaction was purified by HPLC using a Luna Column C-18 (10 µm, 250 mm × 10 mm), with MeOH/H_2_O (85:15) as eluent (flow rate 3.00 mL/min), affording compound **13** (t_R_ = 11.5 min). ^1^H NMR (400 MHz, CD_3_OD): δ_H_ 8.23 (2H, d, *J* = 8.5 Hz), 8.03 (1H, d, *J* = 7.5 Hz), 7.77 (2H, d, *J* = 8.5 Hz), 7.72 (1H, d, *J* = 8.0 Hz), 7.57 (2H, d, *J* = 8.5 Hz), 7.53 (1H, d, *J* = 3.1 Hz), 7.33 (1H, t, *J* = 7.8 Hz), 7.30 (1H, d, *J* = 3.1 Hz), 6.90 (2H, dd, *J* = 8.5 Hz); ^13^C NMR (100 MHz, CD_3_OD): δ_C_ 178.8, 170.2, 159.4, 145.7, 139.0, 133.0, 129.6 (2C), 129.3 (2C), 129.1, 128.2 (2C), 127.8, 126.8, 122.5, 122.4, 117.8, 117.2 (2C), 116.4, 103.6. ESI-MS *m/z* 354.1 [M + H]^+^.

**3-(3′-hydroxy-[1,1′-biphenyl]-4-yl)-5-(*1H*-indol-4-yl)-1,2,4-oxadiazole (14).** An analytic sample of crude reaction was purified by HPLC using a Luna Column C-18 (10 µm, 250 mm × 10 mm), with MeOH/H_2_O (85:15) as eluent (flow rate 3.00 mL/min), affording compound **14** (t_R_ = 12.0 min). ^1^H NMR (700 MHz, CD_3_OD): δ_H_ 8.22 (2H, d, *J* = 8.2 Hz), 8.01 (1H, d, *J* = 7.3 Hz), 7.75 (2H, d, *J* = 8.2 Hz), 7.70 (1H, d, *J* = 8.0 Hz), 7.51 (1H, d, *J* = 3.0 Hz), 7.28 (3H, ovl), 7.13 (1H, d, *J* = 3.0 Hz), 7.12 (1H, d, *J* = 1.8 Hz), 6.82 (1H, dd, *J* = 8.0, 1.8 Hz); ^13^C NMR (175 MHz, CD_3_OD): δ_C_ 178.3, 169.6, 159.5, 145.3, 142.8, 138.5, 131.0, 128.8 (2C), 128.6, 128.4 (2C), 127.4, 127.3, 122.1, 122.0, 119.1, 117.4, 116.2, 115.9, 115.0, 103.4. ESI-MS *m/z* 354.1 [M + H]^+^.

**3-(3′-hydroxy-[1,1′-biphenyl]-3-yl)-5-(*1H*-indol-4-yl)-1,2,4-oxadiazole (15).** An analytic sample of crude reaction was purified by HPLC using a Luna Column C-18 (10 µm, 250 mm × 10 mm), with MeOH/H_2_O (85:15) as eluent (flow rate 3.00 mL/min), affording compound **15** (t_R_ = 12.0 min). ^1^H NMR (700 MHz, CD_3_OD): δ_H_ 8.39 (1H, s), 8.12 (1H, d, *J* = 7.7 Hz), 8.02 (1H, d, *J* = 7.5 Hz), 7.77 (1H, d, *J* = 7.9 Hz), 7.70 (1H, d, *J* = 7.9 Hz), 7.59 (1H, t, *J* = 7.7 Hz), 7.51 (1H, d, *J* = 3.0 Hz), 7.30 (1H, t, *J* = 7.7 Hz), 7.29 (1H, d, *J* = 3.0 Hz), 7.24 (1H, d, *J* = 7.9 Hz), 7.09 (1H, d, *J* = 1.8 Hz), 7.04 (1H, d, *J* = 7.5 Hz), 6.78 (1H, dd, *J* = 8.0, 1.8 Hz); ^13^C NMR (175 MHz, CD_3_OD): δ_C_ 178.1, 169.7, 161.9, 143.6, 142.4, 138.3, 130.6, 130.5, 130.1, 128.6, 128.4, 127.1, 126.6, 126.5, 121.9, 121.7, 117.3, 117.1, 116.8, 116.0, 115.6, 103.1. ESI-MS *m/z* 354.1 [M + H]^+^.

**5-(*1H*-indol-4-yl)-3-(4-((4-methoxycarbonylbenzyl)oxy)phenyl)-1,2,4-oxadiazole (16).** An analytic sample of crude reaction was purified by HPLC using a Luna Column C-18 (10 µm, 250 mm × 10 mm), with MeOH/H_2_O (88:12) as eluent (flow rate 3.00 mL/min), affording compound **16** (t_R_ = 14.0 min). ^1^H NMR (400 MHz, CDCl_3_): δ_H_ 8.47 (1H, br s, NH), 8.21 (2H, d, *J* = 8.8 Hz), 8.10 (3H, ovl), 7.66 (1H, d, *J* = 8.0 Hz), 7.55 (2H, d, *J* = 8.1 Hz), 7.46 (2H, d, *J* = 2.2 Hz), 7.36 (1H, t, *J* = 8.0 Hz), 7.12 (2H, d, *J* = 8.8 Hz), 5.23 (2H, s), 3.95 (3H, s); ^13^C NMR (175 MHz, CDCl_3_): δ_C_ 176.1, 168.3, 166.8, 160.6, 141.7, 136.5, 130.0 (2C), 129.8, 129.2 (2C), 127.0 (2C), 126.6, 126.0, 121.9, 121.8, 120.4, 115.8, 115.6, 115.1 (2C), 103.6, 69.6, 52.4. ESI-MS *m/z* 426.1 [M + H]^+^.

**5-(*1H*-indol-4-yl)-3-(4-((3-methoxycarbonylbenzyl)oxy)phenyl)-1,2,4-oxadiazole (17).** The crude reaction was purified by flash chromatography using a gradient from 9:1 to 8:2 Hexane/Ethyl Acetate affording 30 mg of compound **17** (54% yield). ^1^H NMR (400 MHz, CDCl_3_): δ_H_ 8.54 (1H, s, NH), 8.22 (2H, d, *J* = 8.7 Hz), 8.17 (1H, s), 8.12 (1H, d, *J* = 7.6 Hz), 8.06 (1H, d, *J* = 7.6 Hz), 7.70 (2H, ovl), 7.52 (2H, ovl), 7.38 (1H, t, *J* = 7.6 Hz), 7.30 (1H, s), 7.13 (2H, d, *J* = 8.7 Hz), 5.22 (2H, s), 3.96 (3H, s); ^13^C NMR (100 MHz, CDCl_3_): δ_C_ 175.5, 168.4, 166.3, 160.5, 137.0, 136.5, 131.9, 130.6, 129.4, 129.2 (2C), 128.7, 128.6, 126.6, 126.0, 121.8 (2C), 120.4, 115.8, 115.5, 115.1 (2C), 103.8, 69.3, 52.1. ESI-MS *m/z* 426.1 [M + H]^+^.

### 4.3. Antibiotics and Strains

Vancomycin and oxacillin were purchased from Sigma-Aldrich (Milan, Italy). *Staphylococcus aureus* ATCC 43300 *(*a methicillin-resistant strain characterized by the presence of the *mec* operon), *S. aureus* ATCC 29213, *P. aeruginosa* ATCC 27853, and *K. pneumoniae* ATCC 13883 were obtained from the American Type Culture Collection (Rockville, MD, USA).

### 4.4. Antimicrobial Susceptibility Testing

The minimal inhibitory concentrations (MIC) of all the compounds were determined in Mueller–Hinton medium (MH) by the broth microdilution assay, following the procedure already described [29]. The compounds were added to bacterial suspension in each well, yielding a final cell concentration of 1 × 10^6^ CFU/mL, and a final compound concentration ranging from 1.56 to 100 μM. Compounds **3** and **12** were further tested at the final compound concentrations of 1, 2, 4, and 8 μM. Negative control wells were set to contain bacteria in Mueller–Hinton broth plus the amount of vehicle (DMSO) used to dilute each compound. Positive controls included vancomycin (2 μg/mL) and oxacillin (2 and 10 μg/mL). The MIC was defined as the lowest concentration of the drug that caused a total inhibition of microbial growth after a 24 h incubation time at 37 °C. Medium turbidity was measured by a microtiter plate reader (Biorad mod 680, Milan, Italy) at 595 nm. Minimum bactericidal concentration (MBC) was defined as the lowest test concentration that kills the organism (exhibited no growth on agar plates) after 24 h of incubation at 37 °C.

### 4.5. Killing Rate

Bacterial suspension (10^5^ CFU/mL) was added to the microplates along with compound **12** at the MIC value. The plates were incubated at 37 °C on an orbital shaker at 120 rpm. Viability assessments were performed at 0, 2, 4, 6, and 24 h by plating 0.01 mL undiluted, and 10-fold serially diluted, samples onto Mueller–Hinton plates in triplicate. After the overnight incubation at 37 °C, the bacterial colonies were counted and compared with the counts from the control cultures [30].

### 4.6. Checkerboard Method

The interaction between compound **12** and oxacillin against MRSA was evaluated by the checkerboard method in 96-well microtiter plates containing Mueller–Hinton broth. Briefly, compound **12** and oxacillin were serially diluted along the y and x axes, respectively. The final concentration ranged from 0.03 to 10 µg/mL for oxacillin, and from 0.5 to 3.12 µM (0.5, 0.78, 1, 1.56, 2, 3.12 µM) for **12**. The checkerboard plates were inoculated with bacteria at an approximate concentration of 10^5^ × CFU/mL, and incubated at 37 °C for 24 h, following which the bacterial growth was assessed visually and the turbidity measured by microplate reader at 595 nm. The FIC index for each combination was calculated as follows: FIC index = FIC of **12** + FIC of oxacillin, where FIC of **12** (or oxacillin) was defined as the ratio of the MIC of **12** (or oxacillin) in combination, and the MIC of **12** (or oxacillin) alone. The FIC index values were interpreted as follows: ≤0.5, synergistic; >0.5 to <2.0, additive; >1.0 to ≤2.0, indifferent; and >2.0, antagonistic effects [31].

### 4.7. Molecular Analysis

RNA extraction was performed by using the GenUp Total RNA kit (BiotechRabbit), according to the manufacturer’s instructions. Five hundred nanograms of total cellular RNA were reverse-transcribed (RevertUP II Reverse Transcriptase, BiotechRabbit) into cDNA, using random hexamer primers (Random hexamer, Roche Diagnostics, Germany), at 48 °C for 60 min, according to the manufacturer’s instructions. RT-PCR was carried out using: 2 μL of cDNA amplified in a reaction mixture containing 10 mM Tris–HCl (pH 8.3); 1.5 mM MgCl_2_; 50 mM KCl; 10 µM dNTP; 10 µM forward and reverse primers (*mecA, mec1, mecR1*), or 1 µM forward and reverse 16S rRNA primers; and 2.5 U of Taq DNA polymerase (BiotechRabbit) in a final volume of 25 µL. The cycling conditions are reported in Table 2. The reaction was carried out in a DNA thermal cycler (MyCycler, Biorad, USA). The PCR products were analyzed by electrophoresis on 1.8% agarose gel in TBE, and analyzed on a Gel Doc EZ System (BioRad). Quantification data were normalized to the reference gene for the 16S rRNA gene and analyzed by Image Lab software 5.2.1 (BioRad).

### 4.8. Bioscreens In Vitro for Cytotoxicity Studies

For the cytotoxicity studies, human HaCaT keratinocytes were grown in DMEM (Invitrogen, Paisley, UK), supplemented with 10% fetal bovine serum (FBS, Cambrex, Verviers, Belgium), L-glutamine (2 mM, Sigma, Milan, Italy), penicillin (100 units/mL, Sigma), and streptomycin (100 μ g/mL, Sigma), and cultured in a humidified 5% carbon dioxide atmosphere at 37 °C, according to ATCC recommendations. The cells were inoculated and allowed to grow for 24 h in 96-microwell culture plates at a density of 10^4^ cells/well. The medium was then replaced with fresh medium and the cells were treated for a further 48 h with a range of concentrations (1 → 400 μM) of 12. Using the same experimental procedure, the cell cultures were also incubated with vehicle DMSO as negative control. The cytotoxic activity of compound **12** was investigated through the estimation of a “cell survival index”, arising from the combination of the cell viability evaluation with cell counting [32]. Cell viability was evaluated using the MTT assay procedure, which measures the level of mitochondrial dehydrogenase activity using yellow 3-(4,5-dimethyl-2-thiazolyl)-2,5-diphenyl-2*H*-tetrazolium bromide (MTT, Sigma) as substrate. Briefly, after the treatments, the medium was removed, and the cells were incubated with 20 μL/well of an MTT solution (5 mg/mL) for 1 h in a humidified 5% CO2 incubator at 37 °C. The incubation was stopped by removing the MTT solution, and by adding 100 μL/well of DMSO to solubilize the obtained formazan. Finally, the absorbance was monitored at 550 nm using a microplate reader (iMark microplate reader, Bio-Rad, Milan, Italy). The cell number was determined by a TC20 automated cell counter (Bio-Rad, Milan, Italy), providing an accurate and reproducible total count of the cells, and a live/dead ratio in one step by a specific dye (trypan blue) exclusion assay. Bio-Rad’s TC20 automated cell counter uses disposable slides and TC20 trypan blue dye (0.4% trypan blue dye w/v in 0.81% sodium chloride and 0.06% potassium phosphate dibasic solution). Once the loaded slide is inserted into the slide port, the TC20 automatically focuses on the cells and detects the presence of the trypan blue dye, providing the count. The calculation of the concentration required to inhibit the net increase in the cell number and viability by 50% (IC_50_) is based on plots of data (n = 6 for each experiment) and repeated five times (total n = 30). IC_50_ values were obtained by means of a concentration response curve by nonlinear regression using a curve fitting program, GraphPad Prism 8.0, and are expressed as the mean values ± SEM (n = 30) of the five independent experiments.

## 5. Conclusions

In summary, we have described the discovery of new oxadiazole derivatives as antibiotics active against methicillin-resistant *Staphylococcus aureus.* This study has resulted in the identification of compounds **3** and **12** as the most active of the series. Moreover, we demonstrated that compound **12** works like an antibiotic resistance breaker (ARB), restoring the activity of oxacillin inducing the expression of the genes included in the *mec* operon.

This compound can increase the effectiveness of β-lactam antibiotics by combatting the resistance mechanisms employed against them.

## Data Availability

All data are included in the article or uploaded as Appendix A.

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
