# Peer review of "Synergism of a Novel 1,2,4-oxadiazole-containing Derivative with Oxacillin against Methicillin-Resistant Staphylococcus aureus"

_antibiotics, 2021, doi:10.3390/antibiotics10101258_

Round 1

Reviewer 1 Report

Reviewer comments

This manuscript describes “Synergism of a novel 1,2,4-oxadiazole-containing derivative 2 with oxacillin against methicillin-resistant Staphylococcus aureus”. This is interesting work on 1,2,4-oxadiazole-containing derivatives against MRSA. This class of compounds exhibited interesting activity profile with also exhibited synergism with oxacillin. This is useful work and can be consider for publication. Still, there are few shortcomings that will preclude its publication in the current form.

minor concerns:

  1. Authors need to provide some information about reason ad conclusion of the studies performed. For time kill kinetics and synergistic effect study, author can follow this patter and cite this as reference. (J. Med. Chem. 2017, 60, 15, 6607–6621.)
  2. Results of Time-kill assay of compound 12 against S. aureus ATCC 29213 and S. aureus ATCC 43300 is not clear from table. It would be better if author can add this data in graphical from as well (please see; J. Med. Chem. 2017, 60, 15, 6607–6621 ).
  3. References have few mistakes and not uniform. So authors needs to check all references, i.e. Correct reference 13.
  4. Author can also provide mass spectra and HPLC chromatograms of compounds 3 and 12 in supporting information.

Manuscript can be considered for publication after these corrections.

Author Response

Comments:

  • Authors need to provide some information about reason and conclusion of the studies performed. For time kill kinetics and synergistic effect study, author can follow this patter and cite this as reference. (J. Med. Chem. 2017, 60, 15, 6607–6621.)

Reply. We acknowledge Reviewer1’s comment. In our opinion the rationale and the conclusion of our study are adequately described in the manuscript.

The revised manuscript includes the new reference for time kill kinetics and synergistic effect study (reference 18).

  • Results of Time-kill assay of compound 12 against S. aureus ATCC 29213 and S. aureus ATCC 43300 is not clear from table. It would be better if author can add this data in graphical from as well (please see; J. Med. Chem. 2017, 60, 15, 6607–6621).

Reply. Thank you for the comment. We have reported data in graphical form.

  • References have few mistakes and not uniform. So authors needs to check all references, i.e. Correct reference 13.

Reply. Amended

  • Author can also provide mass spectra and HPLC chromatograms of compounds 3 and 12 in supporting information.

Reply. We have added mass spectra and chromatograms of compounds 3 and 12 in the supplementary file

Reviewer 2 Report

Comments and Suggestions for Authors

In this manuscript, the authors evaluate the antibacterial activity of oxadiazole derivatives against Gram positive and Gram-negative bacteria including MRSA. Of these, the compounds 3 and 12 were most active. The compound 12 inhibited the growth of MRSA and restored the activity of oxacillin. Moreover, combined use of compound 12 and oxacillin showed synergetic effect against MRSA. This is an important contribution of the author in the area of drugs design for multi-drug resistant bacteria. The manuscript is importantly interesting for the readers and meets the requirement of the journal.

I would suggest authors to address following comments/quires before publishing.

Abstract

  1. Please add toxicity data.

Introduction

  1. Line 36. check grammer
  2. Line 55-57 Aim needs to be more clear

Material and Methods

  1. Line 286.Authors mentioned hydroxylamine hydrochloride or different compounds. Correct the name of the chemicals.
  2. Line 319, 327, 335. In the title of the compunds 1H , H should eb italic. Correct it for entire title of compounds.
  3. Section 4.4. Minimum bactericidal concentration (MBC) of drug is usually taken as killing of bacteria by ³99%. Only ³3 log10 killing compared to initial inoculum (106) give ³5 0% killing. Please explain this.

  1. Combine section 4.8 & 4.9
  2. Update the reference number 30

Results

  1. Line 79 and 80 spelling mistakes
  2. Give the abbreviation cfr
  3. Line 102. Rewrite the sentence
  4. Line 124. Rewrite the sentence.
  5. Line 138-138 Rewrite the sentence.
  6. Table 2. I would recommend to present data as graph (time versus log10 etc) for better understanding

Discussion

  1. Please explain bit more about structure activity relationship of the oxadiazole derivatives (especially 3 and 12).
  2. Oxadiazole derivatives gain resistance in bacteria?

Conclusions

  1. No comments

References

  1. No comments

Author Response

Comments:

  1. Abstract: Please add toxicity data.

Reply. Done

  1. Introduction
  • Line 36. check grammar
  • Line 55-57 Aim needs to be more clear

Reply. Done

  1. Material and Methods
  • Line 286.Authors mentioned hydroxylamine hydrochloride or different compounds. Correct the name of the chemicals.
  • Line 319, 327, 335. In the title of the compunds 1H , H should eb italic. Correct it for entire title of compounds.

Reply. Done

  • Section 4.4. Minimum bactericidal concentration (MBC) of drug is usually taken as killing of bacteria by ³99%. Only ³3 log10 killing compared to initial inoculum (106) give ³5 0% killing. Please explain this.

Reply.I’m sorry for the misunderstanding, maybe caused by the definition not appropriate in the results section. We corrected it in the text, but also in material and methods section. We also reported the results in a graph, for their better comprehension. The new figure (Fig 3) better shows the differences between the time 0 (initial inoculum) and 24h of treatment.

  • Combine section 4.8 & 4.9
  • Update the reference number 30

Reply. Done

  1. Results
  • Line 79 and 80 spelling mistakes
  • Give the abbreviation cfr
  • Line 102. Rewrite the sentence
  • Line 124. Rewrite the sentence.
  • Line 138-138 Rewrite the sentence.
  • Table 2. I would recommend to present data as graph (time versus log10 etc) for better understanding

Reply. Thank you for reviewer’s comment. We have modified as suggested.

  • Discussion
  • Please explain bit more about structure activity relationship of the oxadiazole derivatives (especially 3 and 12).

Reply. Thank you for reviewer’s comment. We have modified as suggested, see in discussion section (lines 216-219, and 232-235).

  • Oxadiazole derivatives gain resistance in bacteria?

Reply. Previous investigation by Chang group demonstrated the effectiveness of compound 2 in vivo model of MRSA infection, thus validating the use of this class of oxadiazole derivatives against MRSA infection.